# Contrasting inequality in human exposure to greenspace between cities of Global North and Global South

Bin Chen [1,2,3] ✉, Shengbiao Wu [1], Yimeng Song[4], Chris Webster [2,3,5], Bing Xu[6] ✉ & Peng Gong [2,7] ✉

The United Nations specified the need for "providing universal access to greenspace for urban residents" in the 11th Sustainable Development Goal. Yet, how far we are from this goal remains unclear. Here, we develop a methodology incorporating fine-resolution population and greenspace mappings and use the results for 2020 to elucidate global differences in human exposure to greenspace. We identify a contrasting difference of greenspace exposure between Global South and North cities. Global South cities experience only one third of the greenspace exposure level of Global North cities. Greenspace exposure inequality (Gini: 0.47) in Global South cities is nearly twice that of Global North cities (Gini: 0.27). We quantify that 22% of the spatial disparity is associated with greenspace provision, and 53% is associated with joint effects of greenspace provision and spatial configuration. These findings highlight the need for prioritizing greening policies to mitigate environmental disparity and achieve sustainable development goals.

Greenspace is an important component of urban nature, providing vital ecosystem services to society[1,2] and protecting human health[3–5]. The United Nations specified the need for "providing universal access to greenspace for urban residents" in the 11th Sustainable Development Goal of making cities and human settlements inclusive, safe, resilient and sustainable[6]. However, how far we are from achieving this goal remains unclear in the global context because there are no reliable and accurate data on the exposure of the world's population to nature in the form of green spaces.

Greenspace supply metrics, designed to capture the amount and distribution of green spaces, are widely adopted to gauge the progress toward achieving relevant sustainable and healthy development goals[7,8]. However, measures of provision (in total or per capita) are often simplistically equated to actual exposure on the assumption that the population is evenly exposed to greenspace over time and space[9]. The widely used indicator of greenspace coverage in total does not consider accessibility by the population that might use it[9–11]. Urban greenspace phenology is also rarely considered, which might cause overestimation or underestimation in measurements of greenspace supply. Therefore, aggregated unit measures of greenspace total supply or per capita supply produce an ecological fallacy—biased inferences of individual patterns from aggregated data. Additionally, greenspace privilege is becoming an increasingly critical concern because the inequality in greenspace accessibility has the potential to translate into inequalities in mental and physical health[4,12–14]. Evidence from the United States[15,16], Germany[17,18], Brazil[19], China[10,20] and elsewhere suggests that strong disparities in greenspace supply characterize cities and communities, highlighting the need to consider

[1]Future Urbanity & Sustainable Environment (FUSE) Lab, Division of Landscape Architecture, Department of Architecture, Faculty of Architecture, The University of Hong Kong, Hong Kong SAR, China. [2]Urban Systems Institute, The University of Hong Kong, Hong Kong SAR, China. [3]Musketeers Foundation Institute of Data Science, The University of Hong Kong, Hong Kong SAR, China. [4]School of the Environment, Yale University, New Haven, CT 06511, USA. [5]HKUrbanLabs, Faculty of Architecture, The University of Hong Kong, Hong Kong SAR, China. [6]Department of Earth System Science, Ministry of Education Ecological Field Station for East Asian Migratory Birds, and Institute for Global Change Studies, Tsinghua University, Beijing 100084, China. [7]Department of Geography, and Department of Earth Sciences, The University of Hong Kong, Hong Kong SAR, China. ✉e-mail: binley.chen@hku.hk; bingxu@tsinghua.edu.cn; penggong@hku.hk

better optimization strategies in pursuit of environmental justice and efficiency. Existing studies of greenspace inequality are constrained to the spatial extent of sampled cities[10,15–20] or the measurement scope of greenspace-oriented accessibility statistics[21,22], with limited studies considering the spatially explicit inequality of human exposure to greenspace at the global scale, particularly on the difference in cities between the Global South and North. Moreover, efforts led by local governments or other agencies have often yielded different results with unexplainable differences because of inconsistent measurement and modeling methods. For reliable comparisons across cities and regions, it is critical to perform greenspace exposure assessments at multiple scales derived from the same or consistent data sources using the same or compatible mapping and assessment methods.

To address the challenges outlined above, we undertook a global-scale analysis to uncover differences in human exposure to greenspace, specifically at country, state, county, and city levels. This advances on studies that report greenspace supply density without factoring in the density of demand; we aim to make further advancements by measuring at a lens of high spatial resolution and seasonal variation. A human–greenspace demand–supply (or exposure) relationship was captured using 10-m-resolution satellite vegetation mapping and a 100-m-resolution population dataset. We further selected 1028 large cities (i.e., urban areas ≥100 km$^2$) globally to assess urban greenspace exposure, inequality, and the associated drivers. Our presumption was that a systematic variation in population-weighted greenspace supply globally implies variation in the opportunity for people to enjoy the health and recreational benefits afforded by proximity to green environments. To extend our analysis to seasonal inequalities, we investigated the seasonal variation of urban greenspace exposure. Specifically, we addressed the following three questions. (1) What are the differences in human exposure to greenspace across countries, states, and counties in a global context? (2) What are the differences in greenspace exposure level and exposure inequality among global cities and what are the associated drivers? (3) How does vegetation seasonality affect greenspace exposure and inequality?

## Results

We leveraged fine-resolution global greenspace and population mapping in 2020 to quantify greenspace exposure across countries (Fig. 1a), states (Fig. 1b), and counties (Fig. 1c). Results reveal a prominent spatial difference in the magnitude of greenspace exposure level at all three scales. Our greenspace exposure index is measured as the averaged amount of greenspace coverage within people's nearby environment expressed as a percentage; 100% (best) indicates full greenspace coverage, and 0% (worst) indicates no greenspace coverage within people's nearby living environment. At the country level, we find 45.8% and 21.0% of countries and regions have human exposure to greenspace index values of less than 50% and 25%, respectively (Fig. 1a). The averaged levels of human exposure to greenspace for the top 10 populated countries are 26.5% (China), 24.3% (India), 58.0% (US), 58.7% (Indonesia), 12.2% (Pakistan), 39.4% (Brazil), 47.2% (Nigeria), 35.4% (Bangladesh), 54.0% (Russia), and 28.6% (Mexico). The state-level assessment shows 41.6% and 17.0% of global states have human exposure to greenspace index values of less than 50% and 25%, respectively (Fig. 1b). At the county level, 37.2% and 15.9% of global counties have greenspace exposure index values of less than 50% and 25%, respectively (Fig. 1c). By comparing greenspace exposure level with physical greenspace coverage regardless of the proximal population for each scale, we observe a global "overestimation" problem when considering only greenspace coverage, i.e., our index of human exposure to greenspace is ubiquitously lower than the physical greenspace index (Supplementary Fig. 1). An appropriate buffer zone of greenspace is critical for measuring greenspace exposure. Thus, in addition to the 500-m catchment buffer—widely used for measuring

nearby greenspace exposure [3]—used for our primary analysis (Fig. 1a–c), we also changed the buffer distance to 100, 1000, and 1500 m (Fig. 1d–f). Sensitivity analysis of greenspace exposure ranking shows a consistent pattern of greenspace exposure with different buffer distances; for country-level assessments, the resultant discrepancy between 1500 m and 500 m derived greenspace exposure assessments is slight, i.e., 0.56% on average (Fig. 1d). The corresponding discrepancies are estimated as 0.48% and 2.63% for state- and county-level assessments, respectively (Fig. 1e, f).

We analyzed human exposure to greenspace for the 1028 global cities and found strong global differences of greenspace exposure and inequality (Fig. 2). Global North cities (e.g., US, European, and Australian cities) have higher greenspace exposure (mean: 45.84%) than Global South cities (mean: 14.39%) (e.g., China, India, and the Middle East). Global South cities also experience much higher inequalities in greenspace exposure (Gini: 0.47), i.e., almost twice that of Global North cities (Gini: 0.24). By continent, as shown in Table 1, Asian cities experience the lowest level of human exposure to greenspace (13.49%), which is only one quarter that of North American cities (53.45%), approximately one-third that of European cities (39.25%) and Australian and Oceanian cities (42.51%). Similarly, people in African and South American cities experience greenspace exposure of less than 20% on average, which is much lower than the global mean level of 30.36% (Table 1). The mean Gini index value of greenspace exposure in Asian, African, and South American cities is in the range of 0.41–0.47, which is almost twice that in North American, European, and Australian and Oceanian cities (0.21–0.26), and highlights the urgent need for greenspace planning for health and wellbeing in Global South cities.

Our results also show that urban greenspace exposure inequality is dynamic seasonally (Fig. 3), with the largest discrepancy being, as might be expected, between summer and winter ($R^2 = 0.13$, Fig. 3b, i.e., much lower than that for other paired seasons: $R^2 = 0.22$–$0.71$, Supplementary Fig. 2). Given that vegetation phenological change modifies the temporal availability of urban greenspace (Supplementary Fig. 3) and greenspace exposure (Fig. 3a, d), we tested the Gini index difference of urban greenspace exposure between summer and winter as a function of the spatiotemporal variation of greenspace exposure, as measured by σ/u, and found it to be $R^2 = 0.59$, $p$-value < 0.01 (Fig. 3c).

We incorporated five categories of covariates, including geographic, topographic, climate, landscape, and socioeconomic factors, to examine the drivers of the spatial disparity in greenspace exposure among cities globally. Our statistical analysis reveals that the geographic variable of latitude, climate variables of precipitation and vapor pressure deficit, and landscape variables of greenspace coverage rate and greenspace edge density have a significant association with the Gini index of greenspace exposure (Table 2 and Supplementary Table 1, $p$-value < 0.001). Specifically, greenspace landscape (i.e., greenspace provision and landscape characteristics) accounts for most of the variation of urban greenspace exposure inequality among cities globally ($R^2 = 79.0\%$, Table 2, Supplementary Fig. 4). Greenspace coverage rate, as a measurement of urban greenspace provision, contributes 21.93% of the variance (Supplementary Fig. 4). Greenspace edge density, a measure of the configuration of greenspace independent of quantity, has a unique effect of 3.68% (Supplementary Fig. 4). Overall, 53.42% of urban greenspace exposure inequality is explained jointly by greenspace coverage rate and edge density, as a combined effect of greenspace amount and distribution (Supplementary Fig. 4).

## Discussion

Rapid urbanization over the past decades has reshaped the built environment of the world beyond recognition[11,23,24]. Previous studies reported accentuating environmental inequality as urbanization proceeded, including urban greenspace provision[15,16], accessibility[3,17,21,22], and exposure[10] in specific cities, regions, and even the entire globe. For

 

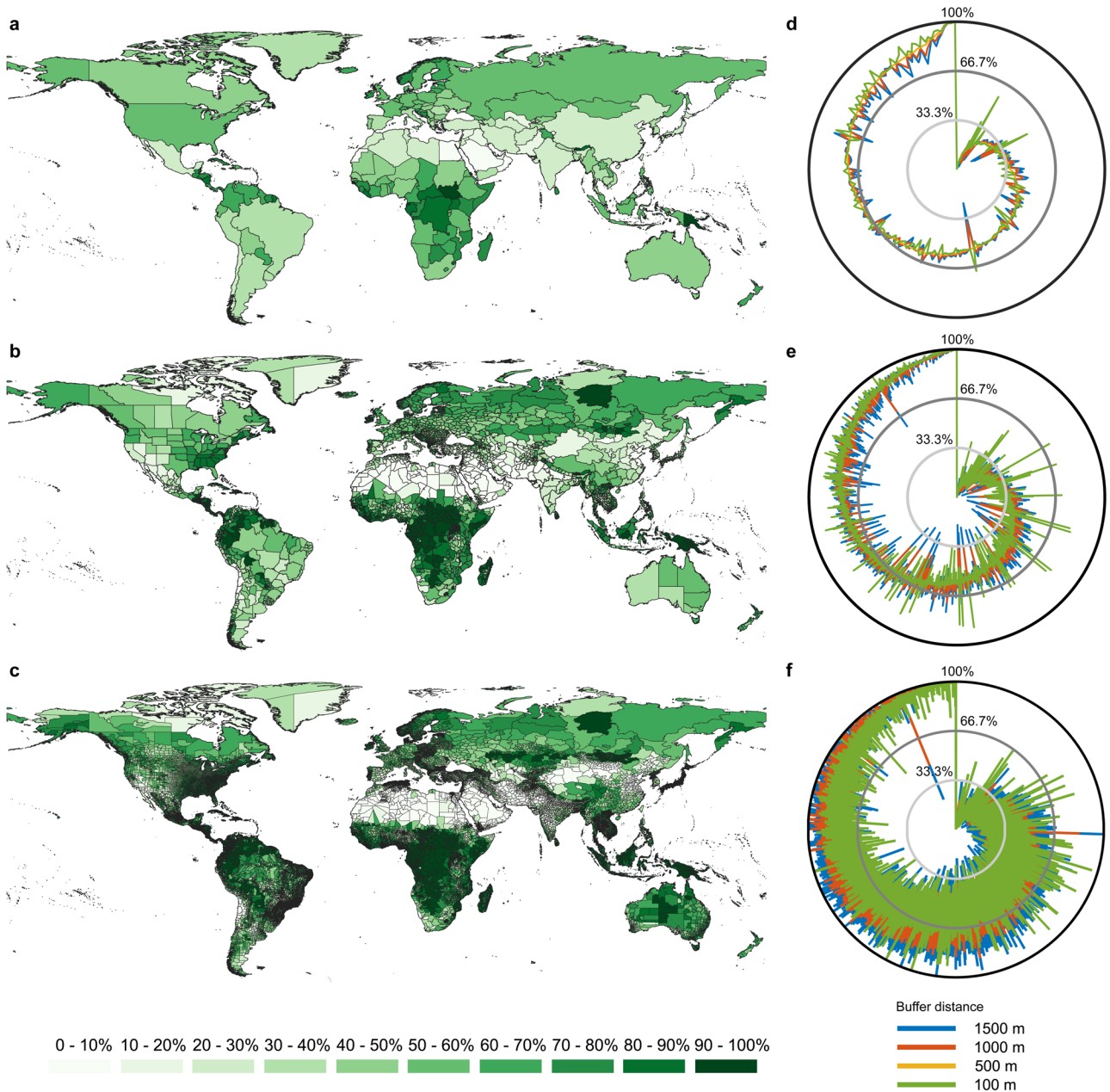

**Fig. 1 | Multiscale heterogeneities of human exposure to greenspace across the globe. a–c** Country-level, state-level, and county-level assessments of human exposure to greenspace using 500 m buffer zones, respectively, with darker green colors showing higher levels of greenspace exposure. Sensitivity of different buffer distances from 100 to 1500 m to the **d** country-level, **e** state-level, and **f** county-level assessments of human exposure to greenspace. The list of countries, states, and counties are ordered by greenspace exposure level estimated using the 500-m buffer zone.

example, Huang et al. (2021) map out the maximum extent of urban greenspace at 30-m spatial resolution and estimate the percentage of the population having a 300-m greenspace accessibility[21]. Similarly, Long et al. (2022) adopt the same accessibility measurement by calculating the proportion of the population that are located inside all 300 m buffer regions of greenspace in a city or country[22]. These methods fall in the scope of greenspace-oriented accessibility statistics. For each unit of greenspace, they estimate the inclusive population within the corresponding buffer regions. However, there are several noticeable shortcomings. First, the accessibility measurement allocates equal greenspace share to a population (i.e., accessible or non-accessible) without differentiating the real amount of greenspace exposed to humans. Second, these methods do not account for all greenspace coverage in urban areas by excluding greenspace areas smaller than a certain size (e.g., the minimum area >0.5 ha or >1 ha).

Given the heterogeneous landscape of cities, certain greenspace types such as street plantation, lawns, and small gardens and parks that play an important role in providing ecosystem services to high-rise and high-density urban areas[25,26] may be omitted. Third, the derived accessibility statistics can represent one-layer reflection of the magnitude in human exposure/accessible to greenspace, but they cannot quantify inequality in a spatially explicit way to account for the share of greenspace benefit for each person. In our study, we extend this research to the global context by characterizing the wall-to-wall fine-resolution footprints of greenspace and population, and mapping the multiscale differences in human exposure to greenspace from country to state, county, and city levels. Based on the results, we further employed the Gini index to quantify the greenspace exposure inequality for global 1028 cities. Our results reveal prominent spatial differences in human exposure to greenspace globally. This

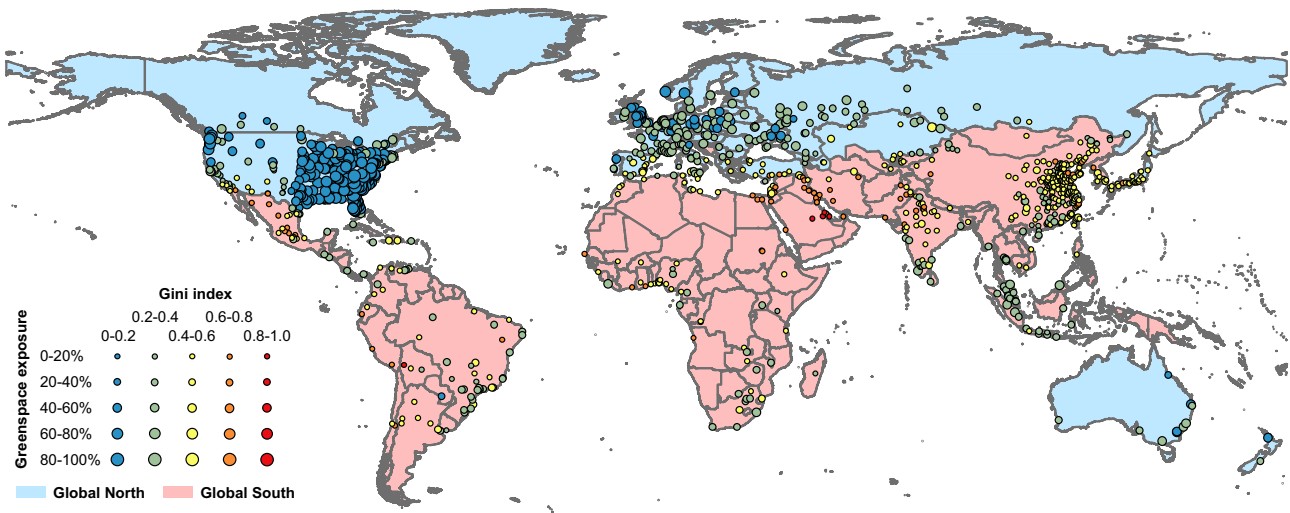

**Fig. 2 | Greenspace exposure levels and the associated greenspace exposure inequalities measured by the Gini index for 1028 cities globally.** Larger bubble sizes represent higher levels of greenspace exposure, and warmer colors represent higher levels of greenspace exposure inequality. Administrative boundaries with light blue (pink) shading represent Global North (South) countries.

heterogeneous phenomenon occurs because of the combined effects of the quantity and distribution of (a) greenspace and (b) population. On the one hand, greenspace coverage will over- or underestimate real greenspace exposure. For example, the greenspace coverage in New York City, United States is 58.01%, but the estimated human exposure to greenspace is only 25.73% (Supplementary Fig. 5a); the greenspace coverage in Billings, United States is 46.79%, but the greenspace exposure is 48.89% (Supplementary Fig. 5b). On the other hand, higher greenspace coverage (supply) of specific administrative units does not guarantee a corresponding higher greenspace exposure (useful supply). The comparison in Supplementary Fig. 5a, b shows that the greenspace coverage in New York City is higher than in Billings, but the real greenspace exposure in Billings is higher than in New York City. Similar findings can also be observed in Global South cities (Supplementary Fig. 5c, d). We observed this "overestimation" at all scales (Supplementary Fig. 1 and Figs. 6–8). The finding, which is consistent with the results of previous studies in China[10], reinforces the importance of considering human–greenspace supply-demand relationships when assessing the adequacy of provision[9].

Our analysis reveals strong contrast between Global South and North cities. Global South cities have only one-third of the greenspace exposure level of Global North cities, and their greenspace exposure inequality is twice that of Global North cities (Table 1). The population of Global North cities in the United States, Canada, European countries, and Australia are much better supplied with greenspace near their living environments, and there is much less unevenness between their cities. Nevertheless, we note that Global North cities have a relatively high standard deviation (Table 1) of greenspace exposure, highlighting the importance of local city planning and policy regarding the supply of this vital class of public good in land markets that would otherwise tend to price out green. For example, out of 180 European cities, 17 are categorized as having "medium inequality" and they are clustered in southern Europe (Fig. 2). Similarly, 23 and 10 out of 293 North American cities are categorized as having "medium inequality" and "high inequality" in greenspace exposure, respectively (Fig. 2). We find a notable association between the Gini index value of urban greenspace exposure and greenspace spatial configuration (greenspace provision and greenspace arrangement) after accounting for other confounding factors. Higher greenspace coverage, as an indicator of greenspace provision in general, tends to reduce greenspace exposure inequality (Table 2, Supplementary Fig. 4). Moreover, different greenspace arrangements in terms of amount and distribution spatially lead to varying greenspace landscapes that have the potential to modify greenspace exposure equality. We note that cities in the North may have a longer history of formal greenspace planning; higher municipal revenues over a century or so to allocate to urban greening; and more mature community feedback systems (including local democracy) to translate demand to supply.

Greenspace exposure level, access, and inequality vary according to season. Many green spaces in high-latitude climates lose some or all their greenness in winter, whereas many green spaces in arid regions lose greenness during dry seasons. Our study leveraging time series remote sensing imagery for seasonal greenspace mapping reveals the magnitude of the seasonal difference in the Gini index (Fig. 3). Because of the different phenological patterns of evergreen and deciduous vegetation across latitudes, the seasonal pattern of urban greenspace exposure also varies between cities of the South and North (Fig. 3a). In winter, some cities have a more stable level of greenspace exposure inequality, while others have a much higher level of greenspace exposure inequality (Fig. 3b). Additionally, human exposure to different greenspace types has been linked with different health benefits[27]. By differentiating the two key components of urban greenspace, i.e., tree (shadowed) and shrub/grass (non-shadowed), we identified some cities distributed in the US–Mexico border region and in North Africa that account for much higher proportions of shrub/grass exposure (Supplementary Figs. 9-10) because of their arid and semiarid climates[28]. Nevertheless, our analysis reveals that tree exposure

**Table 1 | Statistics of city-level human exposure to greenspace and greenspace exposure inequality across regions**

| Region (# of cities) | Greenspace exposure | Gini of greenspace exposure |
|---|---|---|
| Global North (522) | 45.84 ± 20.71% | 0.24 ± 0.13 |
| Global South (506) | 14.39 ± 9.57% | 0.47 ± 0.13 |
| North America (293) | 53.45 ± 22.09% | 0.21 ± 0.14 |
| South America (60) | 19.89 ± 8.09% | 0.41 ± 0.12 |
| Europe (180) | 39.25 ± 11.43% | 0.26 ± 0.08 |
| Africa (63) | 17.66 ± 10.21% | 0.46 ± 0.14 |
| Asia (420) | 13.49 ± 9.52% | 0.47 ± 0.12 |
| Australia/Oceania (12) | 42.51 ± 8.33% | 0.22 ± 0.06 |
| Global (1028) | 30.36 ± 22.58% | 0.35 ± 0.17 |

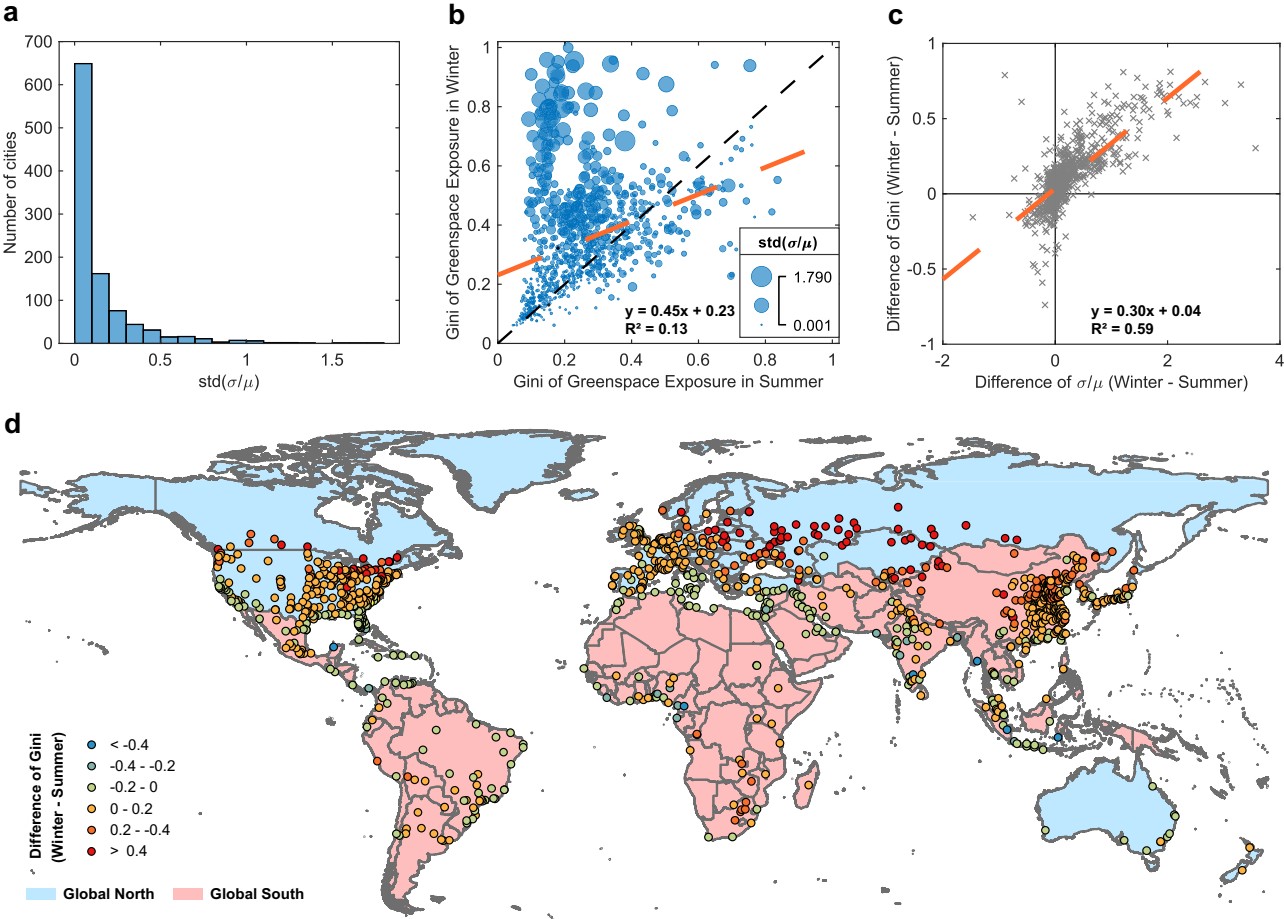

**Fig. 3 | Seasonal change of greenspace exposure inequality for 1028 cities globally.** **a** Histogram of seasonal greenspace exposure variation of each city, measured by the coefficient of spatiotemporal variation – std (σ/u). **b** Difference between the Gini index of greenspace exposure in summer (x-axis) and winter (y-axis). The bubble size represents the magnitude of the spatial and seasonal greenspace exposure variation of each city, measured by std (σ/u). Larger bubble sizes denote higher seasonal variation. **c** Attribution of the Gini index difference of greenspace exposure between summer and winter to the difference of σ/u between summer and winter with a significant positive correlation ($R^2 = 0.59$, p-value < 0.01). **d** Difference between the Gini index of greenspace exposure in summer and winter overlaid on Global South and Global North.

dominates the total greenspace exposure of cities globally (68.66% on average, Supplementary Fig. 9) with high correlation of $R^2 = 0.91$ (Supplementary Fig. 10).

In summary, we found that residents of Global South cities have lower levels of greenspace exposure and that around the lower mean, inequality is higher. This finding is highly relevant regarding both prioritization of greening policies and management actions for mitigating environmental disparity and progress toward achieving sustainable development goals in Global South cities and, ultimately, for the living planet. First, the multi-scale greenspace assessment results

**Table 2 | Summary of multiple linear regression models**

| Category | Covariate | Model 1 | Model 2 | Model 3 | Model 4 |
|---|---|---|---|---|---|
| | Intercept | 0.355*** | 0.355*** | 0.355*** | 0.355*** |
| Geographic | lat | 0.086*** | | | 0.007* |
| Climate | prcp | | −0.029*** | | −0.007** |
| | vpd | | 0.078*** | | 0.027*** |
| Landscape | gcr | | | −0.115*** | −0.104*** |
| | ed | | | −0.047*** | −0.040*** |
| Adjusted R² | | 25.6% | 26.5% | 79.0% | 81.1% |

*lat* latitude; *prcp* precipitation; *vpd* vapor pressure deficit; *gcr* greenspace coverage rate; *ed* greenspace edge density.
***p-value < 0.0001, **p-value < 0.001, *p-value < 0.01.

provide evidence and insights for central and local governments to have a better understanding of the difference between physical greenspace coverage and realistic greenspace exposure, which will help policymakers and planners to implement more effective and sustainable greening programs adjusted to different regional contexts. Second, this study helps detect vulnerable greenspace exposure hotspots in the Global South, which supports local policies and strategies to prioritize improving greenspace supply and creating greener communities, especially in areas of limited and unequal greenspace exposure. Third, our study reveals that greenspace landscape including both provision and spatial configuration, are the main drivers for explaining the variation of urban greenspace exposure inequalities, which calls for coordinated practices among policymakers, city planners, and landscape architects to balance greenspace supply and demand as well as optimize greenspace arrangement for facilitating sustainable and equitable greening management. Specifically, city planners and mayor's offices looking to make their cities healthier through green infrastructure should be wary of aggregate targets, including misleading pro-rata green/person metrics. What matters more is where the green enhancements are located in relation to the urban population. Moreover, it is a fallacy that all kinds of urban greenspace ecosystem services are equally served by a single greenspace plan and configuration. City-scale biophysical and biodiversity benefits may be monitored using aggregate greenspace supply figures. Micro-climate benefits such as targeted urban cool-space design, some

kind of more specific biodiversity benefits, and human health benefits require more attention to how greenspace is configured in shape and proximity to human and urban wildlife populations. We also revealed a seasonal rhythm of urban greenspace exposure inequality and found that the magnitude of changing inequality, as measured by the difference in the Gini index, is highly dependent on the spatiotemporal variation of greenspace exposure. This evidence provides valuable insight for assessing urban green spaces and the associated health benefits by considering both spatial and temporal heterogeneities and represents a quantitative indicator for measuring the bounding range of urban greenspace exposure inequality, as a proxy for the greenspace limit and potential of each city.

Our study is subject to several levels of uncertainty. First, people move during their daily routines, being exposed greenspace environments beyond those near to their place of residence. The spatial distributions of greenspace and population footprints mapped in this study are static at the aggregated level, rather than differentiating spatiotemporal interactions between greenspace and mobile individuals. Nevertheless, we believe that the use of population-weighted models can be interpreted as a useful assessment of human exposure to greenspace for each administrative unit, given that while individuals in a population cluster will travel for work and other purposes, their movements will nevertheless be focused on their place of residence. Our next step is to integrate the human mobility dataset with greenspace observations to derive a spatially and temporally explicit human-greenspace interaction framework and realize a real-time assessment of human exposure to greenspace. Second, differences in population groups are not considered in this study, and we note that different greenspace types, such as trees, grasses, and shrubs, or even different greenspace species of each type will have varying environmental impacts and health benefits for different population groups[27]. Therefore, feeding spatially explicit datasets about demographic attributes such as age, race, income, etc. into the greenspace exposure assessment model can help detect hotspots in shortage of greenspace supply for specific vulnerable population groups and potentially explain the variability of health outcomes. Third, our study achieves a multiscale assessment of greenspace exposure globally for 2020 but does not delineate long-term temporal changes accounting for socioeconomic and climatic changes. The emerging need for research efforts in the next step is to synthesize the past and future for viewing and projecting temporal changes in greenspace exposure and the associated inequalities as a dynamic complex system of evolving interactions among urbanization, climate change, and human interventions. These represent an open topic for further study in this field. Despite the uncertainty inherent in our modeling, our analysis provides a holistic understanding of human greenspace exposure in the global context, especially regarding the contrasting patterns identified between cities of the Global South and North. Upgrading the low levels of health-facilitating greenspace in the Global South and the poor greenspace-provision areas of the Global North cities, will require decisive government and community action, and our analysis serves as a benchmark against which a wide range of future research, practices, and optimizations could be assessed.

## Methods
### Research design
A flowchart outlining the entire research design was provided in Supplementary Fig. 11. We combined fine-resolution population and greenspace mapping with population-weighted exposure models to explore the spatial differences in human exposure to greenspace globally, specifically at country, state, and county levels. Focusing on global large cities, we conducted a comprehensive assessment and comparison of their urban greenspace exposure and the associated inequality. We further performed statistical analysis to examine the main drivers of the highlighted greenspace exposure inequality. To address the third question posed in the introduction, we leveraged time series greenspace observations to investigate the impact of vegetation growth seasonality on greenspace exposure level and inequality.

### Global hierarchy of administrative unit layers
We used the Global Administrative Unit Layers (GAULs) of the Food and Agriculture Organization of the United Nations at country, state, and county administrative levels in 2015 as hierarchical units for the spatial analysis of greenspace exposure globally. GAULs represent the compilation and dissemination of the best available information on administrative units for all countries in the world, providing a contribution to the standardization of the spatial dataset representing administrative units[29].

### Global urban areas
In addition to GAULs, we quantified greenspace exposure for 1028 urban areas globally. Different from the commonly used administrative boundaries, urban area boundaries were based on the latest Global Urban Boundaries shapefiles for 2018, representing the continuous built-up areas[30]. This dataset was extracted from 30-m-resolution Landsat imagery using a hierarchical approach to improve the homogeneity of built-up areas in urban centers and to maintain the heterogeneity of built-up areas at the urban fringes[30]. To ensure a sufficient sample size for statistical analyses using the Gini index, we selected urban areas with a geographic area of >100 km$^2$, which resulted in a total of 1028 urban areas (i.e., 522 are located in Global North and 506 are located in Global South, Supplementary Fig. 12). This group of global 1028 urban areas is used for city-level analysis. To clarify terms: we refer to 1028 'cities' throughout the manuscript to represent the 1028 continuous built-up areas selected for this part of our analysis.

### Population
We used the WorldPop dataset for 2020 to quantify the spatially explicit distribution of population. WorldPop provides the estimated number of people residing in each 100 × 100 m grid based on a random forest model and a global database of administrative unit-based census information[31], which has much finer spatial resolution and update frequency than other population datasets such as the GWP[32] and LandScan[33].

### Greenspace
We leveraged the European Space Agency's latest global baseline land cover product for 2020 (WorldCover) at 10 m spatial resolution to quantify the spatial distribution of greenspace. The WorldCover map includes 11 different land cover classes with overall accuracy of 75% globally[34]. The joint use of Sentinel-1 and Sentinel-2 satellite data not only enhances the spatial resolution of the WorldCover map to 10 m, but also provides reliable land cover information in areas with persistent cloud cover[34]. We extracted all types of forest, shrub, grass, herbaceous wetland, and mangrove from the WorldCover maps as greenspaces.

### Greenspace coverage
We first aggregated the 10 m WorldCover greenspace binary map to 100 m grids to realize the mean fractional greenspace coverage, to ensure the greenspace data is spatially consistent with the 100 m population data (Supplementary Fig. 13). Then, we calculated the physical greenspace coverage rate by overlapping the derived greenspace fractional map with different GAULs according to Eq. (1):

$$GC = \frac{\sum_{i=1}^{N} G_i}{N} \qquad (1)$$

where $G_i$ represents the fractional greenspace coverage of the $i$-th grid, $N$ is the total number of grids within the specific administrative unit,

and *GC* is the physical greenspace coverage rate for this administrative unit.

## Human exposure to greenspace

We applied the population-weighted exposure model[9,10,35,36] to quantify the spatial interaction between population and greenspace. The population-weighted model is a bottom-up assessment that considers the density and distribution of both population and greenspace by allocating higher weights proportionally to greenspace exposure where more people reside according to Eq. (2):

$$GE^d = \frac{\sum_{i=1}^{N} P_i \times G_i^d}{\sum_{i=1}^{N} P_i} \tag{2}$$

where $P_i$ represents the population of the *i*-th grid, $G_i^d$ represents the fractional greenspace coverage of the *i*-th grid considering nearby green environments with a buffered radius of $d$ (i.e., 500, 1000, and 1500 m in this study), $N$ is the total number of grids within the corresponding administrative unit, and *GE* is the corresponding population-weighted greenspace exposure level.

## Greenspace exposure inequality

We adopted the widely used Gini index metric[37] as a measure to assess the global inequality in greenspace exposure following the approach of Song, et al.[10]. Details of the modeling process are provided in the Supplementary Information. The Gini index ranges from 0 (absolute equality) to 1 (absolute inequality), with a lower value indicating that the amount of greenspace exposure is more even and vice versa.

## Model validation

We used three types of greenspace metric (greenspace fraction, population-weighted greenspace exposure, and the Gini index of greenspace exposure), derived from the classification maps of the Sentinel-2 optical satellite images, as benchmarks to evaluate the accuracy of those greenspace metrics derived from the 10-m-resolution European Space Agency WorldCover map. Four major steps were involved in this task. First, we used a maximum value composite approach to generate the yearly vegetation green metrics across 1028 urban cities globally by selecting the maximum normalized difference vegetation index (NDVI) value on a pixel-by-pixel basis from the clear-sky satellite time series. In addition to the maximum NDVI value, we also recorded the corresponding surface reflectance of blue, green, red, and near-infrared spectral bands, and the normalized difference water index for the following image classification purpose. To minimize the uncertainty of cloud cover and cloud shadows for the NDVI composition, we excluded low-quality pixels through the application of cloud and cloud shadow masks obtained from the Sentinel-2 cloud probability product, which records the pixel-based cloud probability using a machine learning approach (https://developers.google.com/earth-engine/tutorials/community/sentinel-2-s2cloudless). The default cloud mask parameters suggested by the Sentinel Hub services and Sentinel Hub cloud detector were adopted in this cloud and cloud shadow removal process. Second, using the composited Sentinel-2 image, we conducted a vegetation and non-vegetation classification using a random forest machine learning approach with a total of 3607 training samples (749 pixels for vegetation and 2858 for non-vegetation) and 15 decision trees. To assess the accuracy of the Sentinel-2 classification, we randomly generated a total of 20,560 validation samples across 1028 urban cities globally (20 validation samples for each city), and excluded 34 of the validation samples that were encompassed by the cloud or cloud cover masks. Consequently, 20,526 effective validation samples were obtained (Supplementary Fig. 15). The vegetation cover conditions of those validation points were determined by the following two steps. 1) We conducted a linear spectral unmixing with the three endmembers of vegetation, non-vegetation, and water across the selected global urban cities and labeled all validation samples with a vegetation fraction larger than the threshold of 0.60 as vegetation pixels. 2) For the remaining validation samples, we classified them as vegetation or not through visual interpretation of both composited Sentinel-2 and high-resolution Google Earth images. With the classification of validation samples, we evaluated the accuracy of the Sentinel-2 classification results, which revealed satisfactory performance for the four widely used metrics (overall accuracy: 0.95, precision: 0.94, recall: 0.95, and F1-score: 0.95, Supplementary Table 2). We also randomly selected 16 urban cities covering the major continents as samples to demonstrate the visual assessment of the Sentinel-2 classification results (Supplementary Fig. 16), which revealed reasonable spatial patterns of classification when compared with the raw Sentinel-2 images. These independent visual and quantitative evaluations collectively supported the feasibility and robustness of the integration of Sentinel-2 imagery and the random forest classifier for deriving greenspace metrics. Third, we applied the calibrated random forest classifier to the Sentinel-2 images across the global urban cities to classify the images into vegetation and non-vegetation components, and we then calculated the city-level greenspace fraction, population-weighted greenspace exposure, and Gini index of greenspace exposure. Finally, we used these greenspace metrics as references to evaluate the accuracy of the corresponding greenspace metrics extracted from the European Space Agency WorldCover map, which revealed overall acceptable accuracy with the estimated regression slope of close to one and high Pearson's correlation coefficients (Supplementary Fig. 17). The slight difference between the greenspace metrics from the two dataset sources is attributable to the greenspace seasonality that arises from vegetation phenology dynamics.

## Seasonal change in greenspace exposure inequality

To explore the seasonal impact of vegetation phenology on greenspace exposure inequality, we first generated Sentinel-2 image compositions in 2020 using the maximum NDVI value composite approach for global urban areas in spring, summer, autumn, and winter. Specifically, in Northern Hemisphere, spring runs from March to May, summer runs from June to August, autumn runs from September to November, and winter runs from December to February. The Southern Hemisphere is the opposite case, i.e., spring runs from September to November, summer runs from December to February, Autumn runs from March to May, and winter runs from June to August. We then used the random forest models to classify the seasonal composites of the Sentinel-2 images. By modeling the spatial human–greenspace interaction using Eq. (2), we further calculated the Gini index of greenspace exposure in each season. We finally investigated the dynamics in the Gini index of greenspace exposure through pairwise comparison (e.g., summer Gini vs. winter Gini) using scatter plots for the four seasons.

To account for the seasonal difference in urban greenspace exposure inequality, we used a dimensionless measure of dispersion, i.e., the coefficient of spatial variation (*csv*), to quantify the spatial variability of greenspace exposure. This coefficient is defined as the ratio of the standard deviation (SD) value ($\sigma$) to the mean value ($\mu$). The *csv* was calculated based on gridded greenspace exposure for each city:

$$csv = \sigma / \mu \tag{3}$$

We further calculated the SD value over the four seasons to quantify the overall spatial and temporal variation (*cstv*) of greenspace exposure:

$$cstv = std(\{csv_i : i = 1, 2, 3, 4\}) \tag{4}$$

## Drivers of greenspace exposure inequality

We compiled a suite of variables to examine their associations with urban greenspace exposure inequality, as measured by the Gini index of greenspace exposure. The inclusive variables comprised five categories (Supplementary Table 1): (i) geographic variables (latitude and longitude), (ii) topographic variables (elevation and slope), (iii) climate variables (mean monthly precipitation, mean monthly temperature, and mean monthly vapor pressure deficit), (iv) socioeconomic variables (nighttime light intensity per $km^2$ as a proxy of Gross Domestic Product, population density per $km^2$, and road length per $km^2$ within the urban areas), and (v) landscape variables (greenspace coverage rate as a proxy of composition, mean patch size, largest patch size, mean patch perimeter–area ratio, mean patch shape index, and edge density as proxies of configuration). We first rescaled all variables to the range of 0–1. Pearson's correlation was calculated to check the correlations between the Gini index of greenspace exposure and these five-category variables (Supplementary Table 1, Supplementary Fig. 11). We then conducted a partial correlation analysis to further quantify the relationships between the Gini index of greenspace exposure and each explanatory variable by controlling the effects of the other variables (Supplementary Table 1, Supplementary Fig. 18). Based on a set of rules including a significance level of a $p$-value < 0.001, |Pearson's r | > 0.1, and |partial correlation coefficient | >0.1, we refined the five explanatory variables for further analysis, including latitude (lat), precipitation (prcp), vapor pressure deficit (vpd), greenspace coverage rate (gcr), and edge density (ed). We then built four ordinary least squares multiple linear regression models to investigate the association between the variables from the different categories and the Gini index of greenspace exposure. The first model included only the geographic variable of latitude. The second model included only the climatic variables of precipitation and vapor pressure deficit. The third model included only the landscape variables of greenspace coverage rate and edge density. The fourth model included all the variables included in models 1–3. Based on the full model result (model 4), we used the variance inflation factor to measure the multicollinearity among the variables. Results showed that the all-inclusive variables achieved a variance in inflation factor of <4 (Supplementary Fig. 19), demonstrating plausible model performance. The adjusted $R^2$, standardized coefficient, and $p$-values were used to assess the regression performance. In addition, we applied a machine learning algorithm, i.e., the random forest model, to build the association between all 16 explanatory variables and the Gini index of urban greenspace exposure. Variable importance was quantified by the indicators of the increase in mean square error and the increase of node purity (Supplementary Fig. 20).

We further employed variance partitioning using the 'vegan' package of R to quantify the relative variations in the Gini index of urban greenspace exposure, which can be explained by the landscape factors into four fractions: (1) unique effect of greenspace provision (i.e., greenspace coverage rate), (2) unique effect of greenspace configuration (i.e., greenspace edge density), (3) joint effects of greenspace provision and spatial configuration, and (4) unexplained residuals.

### Reporting summary

Further information on research design is available in the Nature Research Reporting Summary linked to this article.

## Data availability

Data used in this study are collected from the following sources: The global hierarchy of administrative unit layers are from the Food and Agriculture Organization of the United Nations (https://data.apps.fao.org); Global urban area boundaries are from FROM-GLC research group of Tsinghua University (http://data.ess.tsinghua.edu.cn); Population dataset is from WorldPop (https://www.worldpop.org); Global

baseline land cover product for 2020 (WorldCover) is from the European Space Agency (https://esa-worldcover.org); Sentinel-2 images for validation purpose are from the Sentinel-2 data Archive in Google Earth Engine (https://earthengine.google.com). The resulting greenspace exposure assessments at country, state, and county scales and the greenspace exposure and inequality assessments for global 1028 cities have been deposited at the following repository: https://datahub.hku.hk/projects/GreenExposure/140290.

## Code availability

All code used to produce the necessary data and results in this study are available in the following repository: https://datahub.hku.hk/projects/GreenExposure/140290.

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

## Acknowledgements

This study was supported by the University of Hong Kong HKU-100 Scholars Fund (to B.C.), Seed Fund for Strategic Interdisciplinary Research Scheme Fund (to B.C.), the Research Grants Council of Hong Kong Early Career Scheme (HKU27600222 to B.C.), Major Program of the National Natural Science Foundation of China (42090015 to P.G., 72091514 to B.X.), National Natural Science Foundation of China (42071400 to P.G.), and Tsinghua-Toyota Joint Research Fund (to B.X.).

## Author contributions

B.C. conceived the research idea. B.C., B.X. and P.G. designed the study. B.C. and S.W. performed the main data analysis. B.C. wrote the manuscript. S.W., Y.S., C.W., B.X., and P.G. reviewed and edited the manuscript.

## Competing interests

The authors declare no competing interests.
