## [Peer Review File · Nature Communications]

Reviewer comments, first round

Reviewer #1 (Remarks to the Author):

This study investigated the global differences in human exposure to greenspace. In my view, this study has two impressive contributions. First, the authors identified a contrasting pattern of greenspace exposure between Global South and North cities. They found that Global South cities enjoy only one third of the greenspace exposure level of Global North cities, and greenspace exposure inequality in Global South cities is nearly twice that of Global North cities. Second, they revealed the seasonal variation of greenspace exposure inequality. Generally, the paper is also well-structured and well-written. The paper can be published before the following suggestions have been fully considered.

1. Discussion: The authors found that "High greenspace coverage does not guarantee a corresponding higher greensapce exposure." It is an interesting finding because we normally think that high greenspace coverage means higher greenspace exposure. But I suggest the authors to show us some specific examples (e.g., visualizing the spatial pattern of greenspace coverage and greenspace exposure at a city scale) to prove this finding. I also notice that another study (Huang et al. 2021) has also revealed the global pattern of greenspace exposure for 1039 cities worldwide, but their results (see Figure 5 in the below reference) indicated that high greenspace coverage means higher greensapce exposure, which is different from that of this study.

Reference

Huang et al. Mapping the maximum extents of urban green spaces in 1039 cities using dense satellite images. Environmental Research Letters. 2021.

2. Discussion (L203-206): I understand that this study is highly relevant regarding greening policies and management actions, and achieving SGDs in Global South cities. However, I am looking forward to see more specific policy implications on improving greenspace exposure in Global South cities.

3. Methods: I cannot understand why the authors aggregated the 10-m WorldCover greenspace map to 100-m grids to realize the mean fractional greensapce coverage. Because after the aggregation, there might be mixed pixels in each 100m grid, and they had to set parameters/thresholds to determine which pixels are or are not greenspace. In my view, it is better to use the original 10m grids for the analysis.

4. Seasonal change in greenspace exposure inequality: It will be more reproducible to show us the details on the specific time/year for acquiring the images of different seasons (spring, summer, autumn, and winter). Besides, has the authors considered the differences of seasons in different regions and cities?

5. Fig.1d and Fig.1e. I cannot understand why the human exposure to greenspace is lower with 1500m buffer distance (blue line) than that with 100m buffer distance (green line)?

6. Fig.3: The correlation is 0.46 is the figure c, but it is 0.47 in the caption. Besides, I think it is more useful to visualize the seasonal variation of global cities on maps as a supplement of the Fig 3b.

7. Fig.S4: I felt confused about this figure. How to get or calculate these numbers? What do they mean? What is the relationship between this figure and Table 2?

8. Fig.S14: A legend should be added for this figure.

Reviewer #2 (Remarks to the Author):

This article makes an important contribution to this field of research and is timely. Authors have conducted in-depth research to identify human exposure to green space, greenspace exposure level and exposure inequality, drivers, and effects of seasonal vegetation on green space exposure and inequality. I have enjoyed reading the article.

I have the following comments.

Line 43 - Please include the name of SDG Goal 11 in the abstract.

Line 28 - '..... between Global South and North cities' should be '.....between Global South and Global North cities'

Line 79-82 - includes three research questions. What are the aim and objectives of this research? The methodology/Research Design section should be included before the Results section. A flow chart/diagram should be included to show the various steps of the research to achieve the objectives of the study and also for presenting the overall methodology clearly to the readers. 'Drivers of greenspace exposure inequality' should be included in the Results section. More details on the usefulness of the data used for analysis should be included.

More information is to be included on the following aspects.

What were the selection criteria for selecting 1028 large cities from the global cities?

What are the characteristics of the cities selected? In the method section, it has been identified as 1028 urban areas. But in the manuscript, 1028 cities have been included. Are 'cities' and 'urban areas' terms considered the same? There are different classifications for urban areas.

How many cities are located in Global South, and how many in Global North?

Which are the countries in Global North and Global South where these cities are located?

The manuscript mentioned that human exposure to green spaces was measured across countries, states, and counties. However, the manuscript and supporting materials do not explain this further and only state that 1028 large cities are included in the analysis. The city scale is only mentioned in the abstract.

At least an example of a city, country and county, one each from Global North and Global South, should be included to explain the outcomes of the analysis within the manuscript.

Some relevant and important figures could move into the manuscript from the supplementary section.

The future research directions in the discussion section should be elaborated further. The limitations should be included in the discussion section of the article.

It is better to avoid writing in the first person, such as 'we', and 'I'.

The structure of the article requires some revision.

I would like to see this article published. These revisions are essential for the article for publication. I wish the authors all the very best wishes for their research.

Response letter

> We appreciate both reviewers for their time spent assessing this manuscript and for their thoughtful and valuable comments and suggestions, which are very helpful in improving our manuscript.

> Below is our point-by-point response to the reviewers' specific comments.

Reviewer #1 (Remarks to the Author):

This study investigated the global differences in human exposure to greenspace. In my view, this study has two impressive contributions. First, the authors identified a contrasting pattern of greenspace exposure between Global South and North cities. They found that Global South cities enjoy only one third of the greenspace exposure level of Global North cities, and greenspace exposure inequality in Global South cities is nearly twice that of Global North cities. Second, they revealed the seasonal variation of greenspace exposure inequality. Generally, the paper is also well-structured and well-written. The paper can be published before the following suggestions have been fully considered.

> Thank you very much for your detailed assessment and the above summary.

1. Discussion: The authors found that "High greenspace coverage does not guarantee a corresponding higher greenspace exposure." It is an interesting finding because we normally think that high greenspace coverage means higher greenspace exposure. But I suggest the authors to show us some specific examples (e.g., visualizing the spatial pattern of greenspace coverage and greenspace exposure at a city scale) to prove this finding. I also notice that another study (Huang et al. 2021) has also revealed the global pattern of greenspace exposure for 1039 cities worldwide, but their results (see Figure 5 in the below reference) indicated that high greenspace coverage means higher greenspace exposure, which is different from that of this study.

Reference:

Huang et al. Mapping the maximum extents of urban green spaces in 1039 cities using dense satellite images. Environmental Research Letters. 2021.

> Thanks for your good suggestion. We have added city-level examples from both the Global North and Global South to compare greenspace coverage and greenspace exposure. Results show that higher greenspace coverage does not lead to higher or lower greenspace exposure levels because of the spatial interaction of human-greenspace distribution. As shown in Fig. R1a-b, greenspace coverage will over- or under-estimate the realistic greenspace exposure. The greenspace coverage in New York City, United States is 58.01%, but the human greenspace exposure is only 25.73% (Fig. 1a); the greenspace coverage in Billings, Montana, is 46.79%, but the greenspace exposure is 48.89% (Fig. 1b). Higher greenspace coverage does not indicate a higher greenspace exposure. By comparing Fig. 1a and 1b, we can find the greenspace coverage in New York City is higher than that in Billings, but the realistic greenspace exposure level in Billings is higher than that in New York City. Similar findings can also be observed in Global South cities (Fig. 1c-d)

Fig. R1. City-level comparison of greenspace coverage and greenspace exposure in (a) New York City, New York, United States and (b) Billings, Montana, United States for the Global North, (c) Bloemfontein, Free State, South Africa and (d) Shiyan, Hubei, China for the Global South. Panels from the left to right represent the spatial maps of greenspace coverage, population, normalized greenspace exposure. It should be clarified that greenspace exposure assessment is conducted at the city scale. In order to derive the spatially explicit map of greenspace exposure, we used the normalized greenspace exposure ($\text{population}_i / \text{mean population} * \text{greenspace coverage}_i$) for visual comparison. By averaging the pixel-level normalized greenspace exposure, we can achieve the overall greenspace exposure assessment at the city level.

The detailed revisions are provided in the main text (Page 9, Lines 168-178) and Supplementary materials.

Following this idea, we also compared the difference between physical greenspace coverage and realistic greenspace exposure at the country, state, and county scales. Results reveal a global “overestimation” phenomenon when considering only greenspace coverage, i.e., our index of human exposure to greenspace is ubiquitously lower than the physical greenspace index (Fig. R2).

Fig. R2. Difference between population-weighted greenspace exposure and greenspace coverage rate across different administrative divisions of (a) country, (b) state, and (c) county.

It should be clarified that the study conducted by Huang et al. (2021) first maps out the maximum extent of urban greenspace coverage at 30-m spatial resolution and then estimates the percentage of the population having 300-m greenspace accessibility using Eq. (R1).

$$UGSA = N_{ACC}/N_{TOTAL} \times 100\% \quad (R1)$$

where N_{ACC} is the sum of urban inhabitants that live in 300-m linear distance to urban greenspace that have a size of 0.5 hm² or above, and N_{TOTAL} is the total number of urban inhabitants in the continuous built-up areas.

Based on this estimation, higher greenspace coverages will incorporate more population having the 300-m accessibility to greenspace in the numerator of Eq. (R1), thus leading to a higher greenspace accessibility. However, this approach did not spatial-explicitly factor in human-greenspace supply-demand relationship. As a result, their results measure a

greenspace supply metric.

Accordingly, we have enriched the Discussion about this point in the revised manuscript (Page 9, Lines 167-178), which is duplicated below.

“Our results reveal prominent spatial differences in human exposure to greenspace globally. This heterogeneous phenomenon occurs because of the combined effects of the quantity and distribution of (a) greenspace and (b) population. On the one hand, greenspace coverage will over- or underestimate real greenspace exposure. For example, the greenspace coverage in New York City, United States is 58.01%, but the estimated human exposure to greenspace is only 25.73% (Fig. S5a); the greenspace coverage in Billings, United States is 46.79%, but the greenspace exposure is 48.89% (Fig. S5b). On the other hand, higher greenspace coverage (supply) of specific administrative units does not guarantee a corresponding higher greenspace exposure (useful supply). The comparison in Fig. S5a-b, shows that the greenspace coverage in New York City is higher than in Billings, but the real greenspace exposure in Billings is higher than in New York City. Similar findings can also be observed in Global South cities (Fig. S5c-d).”

2. Discussion (L203-206): I understand that this study is highly relevant regarding greening policies and management actions, and achieving SGDs in Global South cities. However, I am looking forward to see more specific policy implications on improving greenspace exposure in Global South cities.

> Thanks for your suggestion. We have enhanced the Discussion about specific policy implications on improving greenspace exposure in Global South cities in the revised manuscript (Pages 11-12, Lines 225-246), which is duplicated as follows.

“First, the multi-scale greenspace assessment results provide evidence and insights for central and local governments to have a better understanding of the difference between physical greenspace coverage and realistic greenspace exposure, which will help policymakers and planners to implement more effective and sustainable greening programs adjusted to different regional contexts. Second, this study helps detect vulnerable greenspace exposure risk hotspots in the Global South, which supports local policies and strategies to prioritize improving greenspace supply and creating greener communities, especially in areas of limited and unequal greenspace exposure. Third, our study reveals that greenspace landscape including both provision and spatial configuration are the main drivers for explaining the variation of urban greenspace exposure inequalities, which calls for coordinated practices among policymakers, city planners, and landscape architects to balance greenspace supply and demand as well as optimize greenspace arrangement for facilitating sustainable and equitable greening management. Specifically, city planners and mayor’s offices looking to make their cities healthier through green infrastructure should be wary of aggregate targets, including misleading pro-rata green/person metrics. What matters more is where the green enhancements are located in relation to the urban population. Moreover, it is a fallacy that all kinds of urban greenspace ecosystem services are equally served by a single greenspace plan and configuration. City-scale biophysical and biodiversity benefits may be monitored using aggregate greenspace supply figures. Micro-climate benefits such as targeted urban cool-space design, some kind of more specific biodiversity benefits, and human health benefits require more attention to how greenspace is configured in shape and proximity to human and urban wildlife populations.”

3. Methods: I cannot understand why the authors aggregated the 10-m WorldCover greenspace map to 100-m grids to realize the mean fractional greenspace coverage. Because after the aggregation, there might be mixed pixels in each 100m grid, and they had to set parameters/thresholds to determine which pixels are or are not greenspace. In my view, it is better to use the original 10m grids for the analysis.

> Thanks for your good question. The aggregation of 10-m WorldCover greenspace map to 100-m grids is to ensure that greenspace data is spatially consistent with the 100-m WorldPop data set. Also, it should be clarified that we applied the aggregation to derive the fractional greenspace coverage (0-100%) at 100-m grids rather than setting certain thresholds to generate green and non-green hard classification maps. In this way, this operation did not reduce any fine-granule greenspace information.

Fig. R3. Illustrative diagram of aggregating (a) 10-m WorldCover greenspace map to (b) 100-m fractional greenspace coverage map, to be spatially consistent with (c) 100-m WorldPop population grid data, using the City of San Francisco as an example.

Accordingly, we have clarified this issue in the methods (Page 16, Lines 328-330) and added the illustrative diagram in the Supplementary Materials, which is duplicated as follows.

“Greenspace coverage. We first aggregated the 10-m WorldCover greenspace binary map to 100-m grids to realize the mean fractional greenspace coverage, to ensure the greenspace data is spatially consistent with the 100-m population data (Fig. S13).”

4. Seasonal change in greenspace exposure inequality: It will be more reproducible to show us the details on the specific time/year for acquiring the images of different seasons (spring, summer, autumn, and winter). Besides, has the authors considered the differences of seasons in different regions and cities?

> Thanks for your suggestion. To match the 10-m greenspace from WorldCover land cover data in 2020, we selected Sentinel-2 satellite images in 2020 to explore seasonal impacts of vegetation phenology on greenspace exposure inequality, with a seasonal composite using Maximum NDVI composite and random forest classification approach.

Since the phenology of vegetation development is primarily driven by meteorological variables (e.g., solar radiation, air temperature, and precipitation), we thus selected the seasons according to the meteorological definition, which assumes that seasons begin on the first day of the months that include the equinoxes and solstices. Specifically, in the North Hemisphere, spring runs from March to May, summer runs from June to August, Autumn runs from September to November, and winter runs from December to February. The South Hemisphere will be the opposite case, i.e., spring runs from September to November, summer runs from December to February, Autumn runs from March to May, and winter runs from June to August. We did not dive into differentiating local variation of seasons across regions and cities, since the four-season composites can well quantify the extent of changing phenology.

Accordingly, we have included the specific year and months of Sentinel-2 images for the seasonal change in greenspace exposure inequality analysis (Page 20, Lines 403-410), which is duplicated as follows.

“To explore the seasonal impact of vegetation phenology on greenspace exposure inequality, we first generated Sentinel-2 image compositions in 2020 using the maximum NDVI value composite approach for global urban areas in spring, summer, autumn, and winter. Specifically, in Northern Hemisphere, spring runs from March to May, summer runs

from June to August, autumn runs from September to November, and winter runs from December to February. The Southern Hemisphere is the opposite case, i.e., spring runs from September to November, summer runs from December to February, Autumn runs from March to May, and winter runs from June to August.”

5. Fig.1d and Fig.1e. I cannot understand why the human exposure to greenspace is lower with 1500m buffer distance (blue line) than that with 100m buffer distance (green line)?

> Thanks for your good point. Different buffer distances are used to test the sensitivity of greenspace exposure to different ranges of incorporation of nearby green environments (i.e., the relative spatial distribution between humans and greenspace). As the illustrative diagram in Fig. R4 shows, for Scenario (a), green spaces are more closely distributed to humans, and so the derived greenspace exposure using a smaller buffer will be higher than that using a larger buffer. In contrast, there will be the opposite situation for Scenario (b) (i.e., larger buffers will yield a higher greenspace exposure).

Fig. R4. Contrasting scenarios of greenspace exposure assessment using different buffer distances.

Therefore, if a country/state/county’s greenspace exposure is much higher in smaller buffer distances (100m or 500m) than in larger buffer distances (i.e., 1km or 1.5km), we can interpret this to mean that green spaces in this place are distributed much closer to humans. Similarly, we can deduce that green spaces are distributed more remotely from humans if the greenspace exposure assessment is higher using the larger buffer distances.

Also, it should be clarified that human exposure to greenspace is not consistently lower with 1500m buffer distance than that with 100m buffer distance in Fig. 1d-e-f and the zoomed-in figures in Fig. R5. We can identify the fluctuations of rankings derived from different buffer distances.

Fig. R5. Sensitivity of different buffer distances from 100 to 1500 m to the (d) country-level, (e) state-level, and (f) county-level assessments of human exposure to greenspace. The list of countries, states, and counties are ordered by greenspace exposure level estimated using the 500-m buffer zone.

6. Fig.3: The correlation is 0.46 is the figure c, but it is 0.47 in the caption. Besides, I think it is more useful to visualize the seasonal variation of global cities on maps as a supplement of the Fig 3b.

> Thanks for spotting the typo. It should be 0.46 in the original manuscript. In the submitted manuscript, we have revised the coefficient of spatiotemporal variation to $\text{std}(\sigma/u)$, which is more intuitive to characterize the magnitude of changes in greenspace exposure over space and time. Therefore, the correlation has been updated accordingly with an R-square of 0.59. We have updated the caption in the revised manuscript (Page 26, Lines 575-584).

Very good suggestion on visualizing the seasonal variation. Accordingly, we have mapped the difference in Gini values of greenspace exposure between summer and winter to visualize the spatial difference in the revised manuscript, of which the difference is highly latitudinal. Now figure 3 in the main text is updated as follows.

Fig. 3. Seasonal change of greenspace exposure inequality for urban areas globally. (a) Histogram of seasonal greenspace exposure variation of each city, measured by the coefficient of spatiotemporal variation – $\text{std}(\sigma/u)$. (b) Difference between the Gini index of greenspace exposure in summer (x-axis) and winter (y-axis). The bubble size represents the magnitude of the spatial and seasonal greenspace exposure variation of each city, measured by $\text{std}(u/\sigma)$. Larger bubble sizes denote higher seasonal variation. (c) Attribution of the Gini index difference of greenspace exposure between summer and winter to the difference of σ/u between summer and winter with a significant negative correlation ($R^2 = 0.59$, p -value < 0.01). (d) Difference between the Gini index of greenspace exposure in summer and winter overlaid on Global South and Global North.

7. Fig.S4: I felt confused about this figure. How to get or calculate these numbers? What do they mean? What is the relationship between this figure and Table 2?

> Thanks for your question. Fig. S4 is the result of variance partitioning, which is used to quantify the relative variation in the Gini index of urban greenspace exposure explained by explanatory variables. The number represents the percentage of variation that can be explained by a specific variable. We employed the variance partitioning using the “vegan” package in R.

Table 2 shows the statistical results of four comparative models used to detect the key factors regulating urban greenspace exposure inequalities. As shown in Table 2, we can identify that greenspace landscape accounts for most of the variation of urban greenspace exposure inequality among cities globally ($R^2 = 79.0\%$). Based on this finding, we further applied the variance partitioning to quantify the specific contribution from greenspace provisions and greenspace spatial configurations.

To avoid confusion, we have removed Fig. S4a (i.e., the comparative scenario) and just retained Fig. S4b in the revised manuscript. In this case, the variation of Gini index of urban greenspace exposure can be explained by the landscape factors into four fractions: (1) 21.93% of the unique effect from greenspace provision (i.e., greenspace coverage rate), (2) 3.69% of the unique effect from greenspace configuration (i.e., greenspace edge

density), (3) 53.42% of the joint effects from greenspace provision and spatial configuration, and (4) 20.98% of the unexplained residuals.

8. Fig.S14: A legend should be added for this figure.

> Added. The revised figure is shown below.

Fig. S14. Sensitivity of buffer distance and grid size to the assessment of greenspace exposure inequality, as measured by Gini index for global urban areas. **a**, Gini of greenspace exposure estimated using buffer zones from 100 m to 2000 m with an interval of 100 m. The list of 1028 urban areas are ordered by the Gini of Greenspace exposure measured using the 500-m buffer zone. **b**, Gini of greenspace exposure estimated using grid sizes from 100m, 250 m to 2000 m with an interval of 250 m, without considering buffer zones. The list of 1028 urban areas are ordered by the Gini of Greenspace exposure measured using the 500-m grid size. **c**, The variation of Gini of greenspace exposure to buffer distance. **d**, The variation of Gini of greenspace exposure to grid size.

Reviewer #2 (Remarks to the Author):

This article makes an important contribution to this field of research and is timely. Authors have conducted in-depth research to identify human exposure to green space, greenspace exposure level and exposure inequality, drivers, and effects of seasonal vegetation on green space exposure and inequality. I have enjoyed reading the article.

> Thank you for your detailed review and positive comments. According to your suggestions, we have carefully revised our manuscript to better present the methods, results, and discussion.

I have the following comments.

1. Line 43 - Please include the name of SDG Goal 11 in the abstract.

> Thanks for the suggestion. We have specified the name of SDG Goal 11 in the abstract (Page 2, Line 26), which is duplicated below.

"The United Nations specified the need for "providing universal access to greenspace for urban residents" in the 11th Sustainable Development Goal of sustainable cities and communities".

2. Line 28 –'..... between Global South and North cities' should be '.....between Global South and Global North cities'

> Revised accordingly (Page 2, Line 31). Now it reads.

"We identified a contrasting pattern of greenspace exposure between Global South and Global North cities".

3. Line 79-82 - includes three research questions. What are the aim and objectives of this research?

> Thanks for pointing out this issue. We have specified the objective of this study in the revised manuscript (Page 4, Lines 75-79). Now it reads.

"To address the challenges outlined above, we undertook a global-scale analysis to uncover differences in human exposure to greenspace, specifically at country, state, county, and city levels. This advances on studies that report greenspace supply density without factoring in density of demand; and we aim to make further advancements by measuring at a lens of high spatial resolution and seasonal variation."

Addressing this research goal, we aim to answer the following three specific questions (Pages 4-5, Lines 87-91).

"(1) What are the differences in human exposure to greenspace across countries, states, and counties in a global context? (2) What are the differences in greenspace exposure level and exposure inequality among global cities and what are the associated drivers? (3) How does vegetation seasonality affect greenspace exposure and inequality?"

4. The methodology/Research Design section should be included before the Results section. A flow chart/diagram should be included to show the various steps of the research to achieve the objectives of the study and also for presenting the overall methodology clearly to the readers.

> Thanks for the good suggestion. We placed the Method section after the Discussion according to the Nature Communications guideline. Following your suggestion, we have added the flowchart outlining the entire research design (Page 15, Line 287), which is shown as follows.

Fig. R6. Flowchart of the research design including six major steps: (a) Modeling human-greenspace interaction with the population-weighted exposure models, (b) global differences in human exposure to greenspace at country, state, and county scales, (c) selection of global 1028 cities, (d) assessment of greenspace exposure levels and inequalities in global 1028 cities, (e) analysis of drivers for greenspace exposure inequalities, and (f) seasonal change in greenspace exposure levels and inequalities. Specifically, steps (a-b) are designed to address the 1st research question, steps (c-e) are designed to address the 2nd research question, and steps (c-d) and (f) are designed to address the 3rd research question.

5. 'Drivers of greenspace exposure inequality' should be included in the Results section.

> Thanks. This section has been elaborated in the Results section (Page 8, Lines 146-159), which is duplicated as follows.

"We incorporated five categories of covariates including geographic, topographic, climate, landscape, and socioeconomic factors to examine the drivers of the spatial disparity in greenspace exposure between cities globally. Our statistical analysis reveals that the geographic variable of latitude, climate variables of precipitation and vapor pressure deficit, and landscape variables of greenspace coverage rate and greenspace edge density have significant association with the Gini index of greenspace exposure (Tables 2 and S1, p-value < 0.001). Specifically, greenspace landscape (i.e., greenspace provision and landscape characteristics) accounts for most of the variation of urban greenspace exposure inequality between cities globally (R² = 79.0%, Table 2, Fig. S4). Greenspace coverage rate, as a measurement of urban greenspace provision, contributes 21.93% of the variance (Fig. S4). Greenspace edge density, a measure of the configuration of greenspace independent of quantity, has a unique effect of 3.68% (Fig. S4). Overall, 53.42% of urban greenspace exposure inequality is explained jointly by greenspace coverage rate and edge density, as a combined effect of greenspace amount and distribution (Fig. S4)."

6. More details on the usefulness of the data used for analysis should be included.

> Thanks for your suggestion. For each dataset used in this study, we have provided relevant information regarding its reliability and usefulness in the revised manuscript (Pages 15-20, Lines 296-400).

1) Global hierarchy of administrative unit layer (GAULs). GAULs represent the compilation and dissemination of the best available information on administrative units for all countries in the world, providing a contribution to the standardization of the spatial dataset representing administrative units ²⁵.

2) Global urban areas. Different from the commonly used administrative boundaries, urban area boundaries were based on the latest Global Urban Boundaries shapefiles for 2018, representing the continuous built-up areas ²⁶. This dataset was extracted from 30-m-resolution Landsat imagery using a hierarchical approach to improve the homogeneity of built-up areas in urban centers and to maintain the heterogeneity of built-up areas at the urban fringes ²⁶.

3) Population. We used the WorldPop dataset for 2020 to quantify the spatially explicit distribution of population. WorldPop provides the estimated number of people residing in each 100 × 100 m grid based on a random forest model and a global database of administrative unit-based census information ²⁷, which has much finer spatial resolution and update frequency than other population datasets such as the GWP ²⁸ and LandScan ²⁹.

4) Greenspace. We leveraged the European Space Agency's latest global baseline land cover product for 2020 (WorldCover) at 10-m spatial resolution to quantify the spatial distribution of greenspace. The WorldCover map includes 11 different land cover classes with overall accuracy of 75% globally ³⁰. The joint use of Sentinel-1 and Sentinel-2 satellite data not only enhances the spatial resolution of the WorldCover map to 10 m, but it also provides reliable land cover information in areas with persistent cloud cover ³⁰.

5) Greenspace exposure and inequality. We have provided a very detailed section of model validation to verify the reliability of the methods and results.

7. More information is to be included on the following aspects. What were the selection criteria for selecting 1028 large cities from the global cities?

> The selection of the 1028 large cities is based on the criteria of areas > 100 km². We have specified the criteria of city selection in the Methods section-**Global urban areas**.

"Global urban areas. In addition to GAULs, we quantified greenspace exposure for 1028 urban areas globally. Different from the commonly used administrative boundaries, urban area boundaries were based on the latest Global Urban Boundaries shapefiles for 2018, representing the continuous built-up areas ²⁶. This dataset was extracted from 30-m-resolution Landsat imagery using a hierarchical approach to improve the homogeneity of built-up areas in urban centers and to maintain the heterogeneity of built-up areas at the urban fringes ²⁶. To ensure a sufficient sample size for statistical analyses using the Gini index (where n is the number of ~1 × 1 km pixels within the urban area), we selected urban areas with a geographic area of >100 km², which resulted in a total of 1028 urban areas (i.e., 522 are located in Global North and 506 are located in Global South, Fig. S12). This group of global 1028 urban areas is used for city-level analysis. To clarify terms: we refer to 1028 'cities' throughout the manuscript to represent the 1028 continuous built-up areas selected for this part of our analysis."

9. What are the characteristics of the cities selected? In the method section, it has been identified as 1028 urban areas. But in the manuscript, 1028 cities have been included. Are 'cities' and 'urban areas' terms considered the same? There are different classifications for urban areas.

> Yes, they are the same concept in this study. Different from the commonly used administrative boundaries, urban area boundaries were based on the latest Global Urban Boundaries shapefiles for 2018, representing the continuous built-up areas (Li et al., 2021). This dataset was extracted from 30-m-resolution Landsat imagery using a hierarchical approach to improve the homogeneity of built-up areas in urban centers and to maintain the heterogeneity of built-up areas at the urban fringes.

Accordingly, we have clarified in the method section of the revised manuscript (Page 16, Lines 312-314), which is duplicated as below.

“This group of global 1028 urban areas is used for city-level analysis. To clarify terms: we refer to 1028 ‘cities’ throughout the manuscript to represent the 1028 continuously built-up areas selected for this part of our analysis.”

10. How many cities are located in Global South, and how many in Global North?

> Thanks. Table 1 provides the number of Global South (#506) and Global North (#522) cities, which is duplicated as follows.

Table 1. Statistics of city-level human exposure to greenspace and greenspace exposure inequality across regions

Region (# of cities)	Greenspace exposure	Gini of greenspace exposure
Global North (522)	45.84±20.71%	0.24±0.13
Global South (506)	14.39±9.57%	0.47±0.13
North America (293)	53.45±22.09%	0.21±0.14
South America (60)	19.89±8.09%	0.41±0.12
Europe (180)	39.25±11.43%	0.26±0.08
Africa (63)	17.66±10.21%	0.46±0.14
Asia (420)	13.49±9.52%	0.47±0.12
Australia/Oceania (12)	42.51±8.33%	0.22±0.06
Global (1028)	30.36±22.58%	0.35±0.17

Also, we further explicitly provided the number of cities in Global South and Global North in the methods (Page 16, Lines 311-312), which is duplicated as follows.

“We selected urban areas with a geographic area of >100 km², which resulted in a total of 1028 urban areas (i.e., 522 urban areas are located in Global North and 506 are located in Global South, Fig. S12).”

11. Which are the countries in Global North and Global South where these cities are located?

> There are a lot of 126 countries in the Global North (# of 44) and Global South (# of 82), where the global 1028 cities are located, as shown in Fig. R7.

Fig. R7. Geographic location of global 1028 cities overlaid on the classification of Global North and Global South.

Accordingly, the geographic location of global 1028 cities is provided in Fig. 2 and Fig. 3d of the revised manuscript. In the meantime, we have provided the detailed shapefile containing information about country, city, and the associated greenspace exposure levels and inequalities in the Data availability section.

12. The manuscript mentioned that human exposure to green spaces was measured across countries, states, and counties. However, the manuscript and supporting materials do not explain this further and only state that 1028 large cities are included in the analysis. The city scale is only mentioned in the abstract.

> Thanks for your good point. It should be clarified that this study is conducted in a two-stage research design. First, we measured human exposure to greenspace by integrating fine-resolution human-greenspace mapping with population-weighted exposure models, to elucidate the differences in greenspace exposure across countries, states, and counties in the global context, to address our first research question. Second, we dived into city-scale analysis to investigate what are the differences in greenspace exposure level and exposure inequality among global cities and what are the associated drivers, and the seasonal changes of urban greenspace exposure and inequalities, which are the second and third research questions. Based on your previous Comment 4, we have added the workflow of research design (Fig. R7) and the corresponding descriptions to better clarify this issue (Page 14, Line 287, and Supplementary materials).

13. At least an example of a city, country and county, one each from Global North and Global South, should be included to explain the outcomes of the analysis within the manuscript.

> Thanks for your suggestion. We have added corresponding city-, county-, state-, and country-level examples in the revised manuscript, which is duplicated below.

Fig. R8. City-level comparison of greenspace coverage and greenspace exposure in (a) New York City, New York, United States and (b) Billings, Montana, United States for the Global North, (c) Bloemfontein, Free State, South Africa and (d) Shiyang, Hubei, China for the Global South. Panels from the left to right represent the spatial maps of greenspace coverage, population, normalized greenspace exposure. It should be clarified that greenspace exposure assessment is conducted at the city scale. In order to derive the spatially explicit map of greenspace exposure, we used the normalized greenspace exposure ($\text{population}_i / \text{mean population} * \text{greenspace coverage}_i$) for visual comparison. By averaging the pixel-level normalized greenspace exposure, we can achieve the overall greenspace exposure assessment at the city level.

Fig. R9. Country-level comparison of greenspace coverage and greenspace exposure in

(a) Slovakia (Global North) and (b) Bolivia (Global South).

Fig. R10. State-level comparison of greenspace coverage and greenspace exposure in (a) Banská Bystrica, Slovakia (Global North) and (b) Cochabamba, Bolivia (Global South).

Fig. R11. County-level comparison of greenspace coverage and greenspace exposure in (a) Banská Bystrica, Slovakia (Global North) and (b) Cercado, Cochabamba, Bolivia (Global South).

The detailed revisions are provided in the main text (Page 9, Lines 168-178) and Supplementary materials.

14. Some relevant and important figures could move into the manuscript from the supplementary section.

> Due to the limited display items in *Nature Communications*, we have re-organized the figures in the revised manuscript by some combinations.

15. The future research directions in the discussion section should be elaborated further. The limitations should be included in the discussion section of the article.

> Thanks for the suggestion. The potential uncertainties (limitations) have been acknowledged in the discussion, and we have further elaborated more details about the future research directions in the revised manuscript (Pages 13-14, Lines 254-278), which is duplicated as follows.

“Our study is subject to several levels of uncertainty. First, people move during their daily routines, being exposed greenspace environments beyond those near to their place of residence. The spatial distributions of the greenspace and population footprints mapped in this study are static at the aggregated level, rather than differentiating spatiotemporal interactions between greenspace and mobile individuals. Nevertheless, we believe that the use of population-weighted models can be interpreted as a useful assessment of human exposure to greenspace for each administrative unit, given that while individuals in a population cluster will travel for work and other purposes, their movements will nevertheless be focused on their place of residence. Our next step is to integrate human mobility dataset with greenspace observations to derive spatially and temporally explicit human-greenspace interaction framework and realize real-time assessment of human exposure to greenspace. Second, differences in population groups are not considered in this study, and we note that different greenspace types such as trees, grasses, and shrubs, or even different greenspace species of each type will have varying environmental impacts and health benefits for different population groups 23. Therefore, feeding spatially explicit dataset about demographic attributes such as age, race, income, etc. into the greenspace exposure assessment model can help detect hotspots in shortage of greenspace supply for specific vulnerable population groups, and potentially explain the variability of health outcomes. Third, our study achieves a multiscale assessment of greenspace exposure globally for 2020 but does not delineate long-term temporal changes accounting for socioeconomic and climatic changes. The emerging need for research efforts in the next step is to synthesize the past and future for viewing and projecting temporal changes in greenspace exposure and the associated inequalities as a dynamic complex system of evolving interactions among urbanization, climate change, and human interventions. These represent an open topic for further study in this field.”

16. It is better to avoid writing in the first person, such as 'we', and 'I'.

> Thanks for the suggestion. We have tried to avoid using the first person in the revised manuscript. In the meantime, after reviewing papers from Nature family journals in recent years, we acknowledge that this writing style has been quite acceptable and popular. Below are some examples:

<https://www.nature.com/articles/s41893-022-00872-1>

<https://www.nature.com/articles/s41586-020-3005-2>

<https://www.nature.com/articles/s41467-022-29094-x>

<https://www.nature.com/articles/s41467-022-30146-5>

<https://www.nature.com/articles/s41467-022-30121-0>

17. The structure of the article requires some revision.

> We have strictly followed the guidelines of *Nature Communications* and made some further revision according to your suggestion.

18. I would like to see this article published. These revisions are essential for the article for publication. I wish the authors all the very best wishes for their research.

> Thanks for the encouraging words. We appreciate your assessment and detailed feedback.

Reviewer comments, second round

Reviewer #1 (Remarks to the Author):

The authors have answered most of my comments! But, please be careful to state that "Existing studies of greenspace inequality are typically constrained to a single city or to a sample of cities, with none considering the global scale." To the best of my knowledge, the two references below have already reported the exposure of population to greenspace for even more cities across the globe. Thus these references should be cited and fully discussed because they are highly related to this study. Moreover, I understand that the authors have used a different model to measure population exposure to greenspace. Nevertheless, the difference between using different measures should also be discussed.

- 1.Huang et al. Mapping the maximum extents of urban green spaces in 1039 cities using dense satellite images. Environmental Research Letters. 2021.
- 2.Long et al. Visualizing green space accessibility for more than 4,000 cities across the globe. Environment and Planning B: Urban Analytics and City Science. 2022.

Reviewer #2 (Remarks to the Author):

The authors have made all the required revisions to the revised manuscript following the review comments. New tables, figures and further explanations in the manuscript and supplementary materials are included in the revised version. The manuscript now presents this research clearly. This manuscript can be now accepted for publication. Thank you.

Response letter

> We appreciate both reviewers for their time spent assessing the revised manuscript and for their further comments and suggestions, which are helpful in improving the quality of our manuscript.

> Below is our point-by-point response to the reviewers' specific comments.

Reviewer #1 (Remarks to the Author):

The authors have answered most of my comments! But, please be careful to state that "Existing studies of greenspace inequality are typically constrained to a single city or to a sample of cities, with none considering the global scale." To the best of my knowledge, the two references below have already reported the exposure of population to greenspace for even more cities across the globe. Thus these references should be cited and fully discussed because they are highly related to this study. Moreover, I understand that the authors have used a different model to measure population exposure to greenspace. Nevertheless, the difference between using different measures should also be discussed.

1. Huang et al. Mapping the maximum extents of urban green spaces in 1039 cities using dense satellite images. *Environmental Research Letters*. 2021.
2. Long et al. Visualizing green space accessibility for more than 4,000 cities across the globe. *Environment and Planning B: Urban Analytics and City Science*. 2022.

> Thank you for your detailed assessment on our revised manuscript. We appreciate your comment and suggestion, which are very helpful to further clarify the novelty of this study.

First, according to your suggestion, we have revised the statement in the Introduction to "*Existing studies of greenspace inequality are constrained to the spatial context of sampled cities or the measurement scope of greenspace-oriented accessibility statistics, with limited studies considering the spatially explicit inequality of human exposure to greenspace at the global scale, particularly on the difference in cities between the Global South and North*" (Page 4, Lines 66-70).

Second, our manuscript aims to answer three scientific questions: (1) What are the differences in human exposure to greenspace across countries, states, and counties in the global context? (2) What are the differences in greenspace exposure level and exposure inequality among global cities and what are the associated drivers? (3) How does vegetation seasonality affect greenspace exposure and inequality?

Regarding the raised point that falls into the methodology of measuring greenspace exposure, we have carefully read through the two recommended papers and cited them accordingly (Page 4, Line 68; Page 9, Lines 164-170). It should be clarified that the study conducted by Huang et al. (2021) first maps out the maximum extent of urban greenspace coverage at 30-m spatial resolution and then estimates the percentage of the population having 300-m greenspace accessibility using Eq. (R1),

$$UGSA = N_{ACC}/N_{TOTAL} \times 100\% \quad (R1)$$

where N_{ACC} is the sum of urban inhabitants that live in 300-m linear distance to urban greenspace that have a size of 0.5 hm² or above, and N_{TOTAL} is the total number of urban inhabitants in the continuous built-up areas.

Similarly, the study conducted by Long et al. (2022) also adopts the same accessibility measurement by calculating the proportion of population that are located inside all 300-m buffer regions of greenspace (i.e., area > 1 hm²) in a city or a country. These methods fall in the scope of greenspace-oriented accessibility statistics. For each unit of greenspace, this type of methods will calculate the inclusive population within the corresponding 300-m buffer regions. However, there are several noticeable shortcomings. First, the accessibility measurement allocates equal greenspace share to a population (i.e., accessible or non-accessible) without differentiating the real amount of greenspace exposed to humans. Second, these methods do not account for all greenspace coverage in urban areas. As one can spot from these literatures (e.g., Huang et al. 2021; Long et al. 2022), there will be thresholds (i.e., the minimum area > 0.5 or 1 hm², that is 5,000 or 10,000 m²) used to exclude small greenspace patches. Given the heterogeneous landscape of cities, certain greenspace types such as street plantation, lawns, and small gardens and parks that play an important role in providing ecosystem services to high-rise and high-density urban areas may be omitted. Third, the derived accessibility statistics can represent one-layer reflection of the magnitude in human exposure/accessible to greenspace, but they cannot quantify inequality in a spatially explicit way to account for different greenspace benefit to each people in corresponding cities.

Therefore, our study adopts the population-weighted exposure method to model the spatial interaction between all greenspace and population footprints and derive the estimate of human exposure to greenspace. Based on this result, we further employed the Gini index (as shown in the illustrative diagram in Fig. R1) to quantify the inequality of greenspace exposure and investigate the associated drivers and seasonal changes, which would be a good contribution to existing literatures.

Fig. R1. Illustrative diagram of Gini index-based inequality assessments of

greenspace exposure. The Gini index is defined as the ratio of the area that lies between the line of equality and the Lorenz curve (region A) over the total area under the line of equality (region A plus region B), where Lorenz curve plots the proportion of the greenspace exposure (y-axis) that is cumulatively shared by the residents (x-axis). B_i indicates the contribution of i th residents to the accumulated greenspace exposure and is estimated by the trapezoid area as shown in the left panel, where g_i represents the greenspace that is exposed to i th resident, and n represents the total resident number. Y-axis shows the cumulative share of greenspace exposure; X-axis shows the cumulative share of residents from lowest to highest greenspace exposure.

Accordingly, we have made a detailed discussion in terms of the methods and research scopes in the revised manuscript (Pages 9-10, Lines 163-186), which is duplicated below.

“Previous studies reported accentuating environmental inequality as urbanization proceeded, including urban greenspace provision^{15,16}, accessibility^{3,17,21,22}, and exposure¹⁰ in specific cities, regions, and even the entire globe. For example, Huang et al. (2021) maps out the maximum extent of urban greenspace at 30-m spatial resolution and estimates the percentage of the population having a 300-m greenspace accessibility²¹. Similarly, Long et al. (2022) adopts the same accessibility measurement by calculating the proportion of population that are located inside all 300-m buffer regions of greenspace in a city or country²². These methods fall in the scope of greenspace-oriented accessibility statistics. For each unit of greenspace, they estimate the inclusive population within the corresponding buffer regions. However, there are several noticeable shortcomings. First, the accessibility measurement allocates equal greenspace share to a population (i.e., accessible or non-accessible) without differentiating the real amount of greenspace exposed to humans. Second, these methods do not account for all greenspace coverage in urban areas by excluding greenspace areas smaller than a certain size (e.g., the minimum area > 0.5 ha or > 1 ha). Given the heterogeneous landscape of cities, certain greenspace types such as street plantation, lawns, and small gardens and parks that play an important role in providing ecosystem services to high-rise and high-density urban areas^{25,26} may be omitted. Third, the derived accessibility statistics can represent one-layer reflection of the magnitude in human exposure/accessible to greenspace, but they cannot quantify inequality in a spatially explicit way to account for the share of greenspace benefit for each person. In our study, we extend this research to the global context by characterizing the wall-to-wall fine-resolution footprints of greenspace and population, and mapping the multiscale differences in human exposure to greenspace from country to state, county, and city levels. Based on the results, we further employed the Gini index to quantify the greenspace exposure inequality for global 1028 cities.”

Ref:

Huang et al. Mapping the maximum extents of urban green spaces in 1039 cities using dense satellite images. *Environmental Research Letters*. 2021.

Long et al. Visualizing green space accessibility for more than 4,000 cities across the globe. *Environment and Planning B: Urban Analytics and City Science*. 2022.

Reviewer #2 (Remarks to the Author):

The authors have made all the required revisions to the revised manuscript following the review comments. New tables, figures and further explanations in the manuscript and supplementary materials are included in the revised version. The manuscript now presents this research clearly. This manuscript can be now accepted for publication. Thank you.

> Thank you again for your detailed assessment on our revised manuscript. We appreciate your comments and suggestions during the first round of review, which helped to clarify the research details and improve the quality of this manuscript.